# Detection of false position attacks in VANETs through bagging ensemble learning

**Bekan Kitaw Mekonen** [ID]1*, **Lemi Bane**2*, **Negasa Berhanu Fite** [ID]1

**1** Faculty of Computing and Informatics, Jimma Institute of Technology, Jimma University, Jimma, Oromia, Ethiopia, **2** Department of Computer Science, Dambi Dollo University, Dembi Dolo, Oromia, Ethiopia

* bekan.mekonen@ju.edu.et (BKM); lemibane@dadu.edu.et (LB)

## Abstract

Vehicular Ad-hoc Networks (VANETs) are critical to Intelligent Transportation Systems (ITS), enabling vehicle-to-vehicle (V2V) and vehicle-to-infrastructure (V2I) communication to improve road safety and traffic flow. However, VANETs face significant security threats, particularly position falsification attacks, where malicious nodes disseminate false Basic Safety Messages (BSMs). This study proposes an ensemble learning framework to detect such attacks, leveraging Decision Tree (CART), Random Forest, K-Nearest Neighbors (KNN), and Multilayer Perceptron (MLP) classifiers enhanced with bagging. Using the VeReMi dataset, our RSU-level detection system analyzes sequential BSMs to detect malicious behavior. Results demonstrate that KNN with bagging achieves perfect precision, recall, accuracy, and F1 score (100%) for Attack 1, while maintaining near-perfect performance for complex attacks like Attack 2 (99.87% accuracy) and Attack 16 (97.85% accuracy). Decision Tree with bagging also performs well for simpler attacks but experiences a slight decline for highly complex scenarios. Random Forest with bagging excels in simpler attacks but struggles with complex patterns. MLP with bagging shows strong results for simpler attacks but underperforms in complex scenarios. The proposed framework highlights the effectiveness of ensemble techniques, particularly KNN with bagging, in safeguarding VANET communication systems, offering a scalable, efficient, and robust solution for VANET security.

## 1. Introduction

Vehicular Ad Hoc Networks (VANETs), a specialized type of Mobile Ad Hoc Networks (MANETs), facilitate communication among vehicles (V2V) and between vehicles and infrastructure (V2I) [1], as illustrated in Fig 1. These networks are fundamental to Intelligent Transportation Systems (ITS) [2], enhancing road safety and traffic management while providing comfort and safety to vehicle occupants. With the dynamic nature of VANETs [3], characterized by constantly moving vehicles and frequent changes in network topology, efficient and real-time communication is essential.

**Data availability statement:** All data underlying the findings reported in this study are available in figshare at https://doi.org/10.6084/m9.figshare.29322179 under the Creative Commons Attribution (CC BY) license.

**Funding:** The author(s) received no specific funding for this work.

**Competing interests:** The authors have declared that no competing interests exist.

Delays in information transmission can significantly impact the network's performance and reliability [4,5]. Therefore, ensuring the authenticity, confidentiality, and integrity of information is paramount. To address these security requirements, a robust misbehavior detection model is necessary to safeguard against malicious activities that exploit vulnerabilities in VANET communications, such as the dissemination of false position information. While cryptographic solutions like Public Key Infrastructure (PKI) provide a foundation for secure communication, their limitations in verifying data accuracy necessitate advanced detection mechanisms to counter such threats.

A critical enabler of secure communication in VANETs is the Public Key Infrastructure (PKI), a cryptographic framework designed to ensure the authenticity and integrity of data exchanged within ITS [6]. PKI relies on digital certificates issued by trusted Certificate Authorities (CAs), which bind public keys to vehicle identities, allowing vehicles to sign Basic Safety Messages (BSMs) with private keys. This ensures that V2V and V2I communications are authentic and resistant to impersonation. In ITS, PKI supports critical functions such as secure dissemination of traffic alerts, collision avoidance warnings, and cooperative driving information, enhancing road safety and efficiency. Its advantages include robust authentication, scalability across large-scale VANETs, and compatibility with existing ITS infrastructure, making it a cornerstone of vehicular network security.

However, while PKI ensures message authenticity through cryptographic methods like digital signatures, it cannot validate the accuracy of the transmitted data, such as falsified position coordinates in BSMs [7]. Therefore, ensuring the authenticity, confidentiality, and integrity of information is paramount. While the Public Key Infrastructure (PKI) model [6] addresses authenticity through cryptographic methods like digital signatures, it falls short of ensuring the integrity and accuracy of transmitted data [7]. PKI alone cannot validate the correctness of the information, highlighting the necessity of an additional layer of security. This gap calls for the integration of a misbehavior detection model, one that can ensure message accuracy and safeguard against malicious activities within the VANETs.

Among the various security threats faced by VANETs, position falsification attacks [8] where malicious vehicles disseminate false BSMs, are particularly concerning. In these attacks, an attacker vehicle deliberately broadcasts inaccurate position coordinates within BSMs, misleading legitimate vehicles and compromising the network's integrity. Traditional cryptographic techniques such as rule-based and statistical approaches [9] are insufficient to detect such attacks. Besides, previous studies have utilized machine learning algorithms to detect misbehavior in the network, emphasizing the importance of high accuracy in identifying such behavior. Our research introduces a vehicle-RSU pair-based approach within sequential BSMs, leveraging the capabilities of machine learning such as Decision Tree [10], Random Forest [11], K-Nearest Neighbors (KNN) algorithms [12], and MLP classifier, each enhanced with bagging techniques [13] to enhance detection accuracy. Decision Tree with Bagging strengthens single-tree classifiers by aggregating predictions through majority voting, reducing overfitting, and improving accuracy. This ensemble framework, grounded in the theoretical strength of combining diverse models to handle noisy VANET

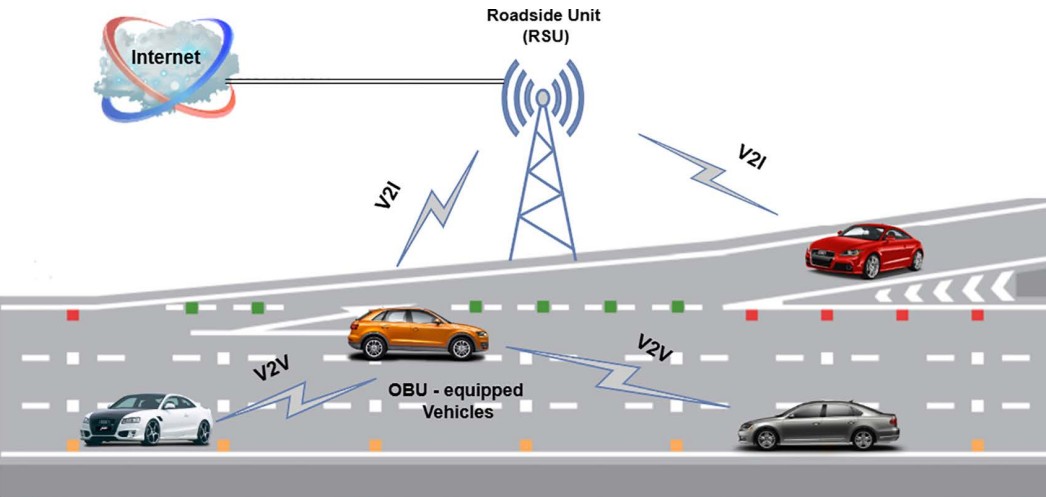

**Fig 1. Communications in VANET [8].**

data, achieves high detection accuracy, with KNN with Bagging emerging as the most effective due to its scalability and resilience.

By employing RSUs as computational units, the framework minimizes the resource burden on vehicles while maintaining high detection accuracy. This approach reduces model variance, mitigates noise, and enhances generalization, resulting in a scalable system for detecting position falsification attacks. The study's innovation lies in integrating bagging with KNN, achieving near-perfect detection metrics, thus significantly improving the security and reliability of VANETs for safer and more efficient transportation networks.

## 2. Rationale

Vehicular Ad-Hoc Networks (VANETs) have emerged as a transformative technology in the realm of intelligent transportation systems, providing significant improvement in road safety, traffic management, and driver comfort. By enabling real-time communication between vehicles (V2V) and infrastructure (V2I), VANETs facilitate the dissemination of critical information such as traffic conditions, hazard warnings, and emergency alerts [14,15]. However, due to their open and dynamic nature, they are highly susceptible to security threats, particularly position falsification attacks, where malicious vehicles disseminate false location information to disrupt network operations or cause accidents.

Conventional security mechanisms, such as Public Key Infrastructure (PKI) [16], ensure message authentication but cannot verify the accuracy of transmitted data, underscoring the need for advanced, data-driven detection strategies. Recent studies highlight machine learning's effectiveness in distinguishing malicious from legitimate behavior. For instance, [17] proposed a data-centric framework using behavioral features (e.g., trajectory deviations, inconsistent speeds) to train classifiers like Naïve Bayes, Random Forest, and Decision Tree via the WEKA toolkit [18]. Their ensemble approach, using majority voting, outperformed single classifiers, with Random Forest and Decision Tree excelling. Similarly, [19] used a predictive framework analyzing vehicle trajectories to detect position falsification, with Random Forest achieving high accuracy by comparing predicted and actual BSM positions.

Complementing these approaches, [20] investigated the impact of feature selection on attack detection. The authors examined three feature combinations: (1) position and speed, (2) position, speed, and coordinate changes, and (3) position, speed, and variation in both speed and coordinates. Their analysis showed that speed alone had a minimal impact on detection accuracy, while incorporating positional dynamics such as sudden coordinate changes significantly improved

performance. They employed Support Vector Machine (SVM) and Logistic Regression models, with SVM showing slight superiority in managing nonlinear feature relationships. Additionally, the authors [21] introduced a trusted neighbor table (TNT) to combat position-spoofing attacks. This location verification method requires each vehicle to maintain a TNT containing the most recent authenticated locations of its nearby vehicles. Unlike conventional neighbor tables, the TNT assigns trust scores to entries, with higher values indicating more reliable neighbors. The authors claim that their approach is both secure and efficient, especially in situations where no supporting infrastructure is present.

In [13], the authors of the VeReMi dataset presented Maat, a framework based on subjective logic to verify the reliability of received data. Maat uses subjective opinions to represent trust and confidence in data, employing belief theory principles similar to the Dempster-Shafer theory. The framework incorporates four comparison checks: Acceptance Range Threshold (ART), Sudden Appearance Warning (SAW), Simple Speed Check (SSC), and Distance Moved Verifier (DMV), to evaluate data reliability. These checks verify transmission ranges, beacon intervals, speed consistency, and movement plausibility, respectively. Building on this, in [22], the authors proposed combining plausibility checks with a machine learning framework for misbehavior detection using the VeReMi dataset. They introduced six features, including location and movement plausibility checks, along with quantitative features that capture velocity and displacement inconsistencies. This approach improves the detection of false locations and anomalous behavior. A misbehavior detection model, named a sender-receiver pair approach [23], was proposed to tackle both false alert verification and position falsification attacks. The framework uses the Green shield model to estimate traffic conditions. Features such as speed, position, receiving distance, and RSSI values are used to detect misbehavior, with results aggregated to identify and evict malicious vehicles from the network.

These studies provide a foundation for misbehavior detection, but our approach advances this by leveraging ensemble learning's robustness for position falsification detection. We adopt a suite of classifiers—Decision Tree, Random Forest, KNN, and MLP—enhanced with bagging to mitigate noise and improve generalization in VANETs' dynamic environment. Unlike single-classifier methods, our ensemble approach, particularly KNN with Bagging, offers superior accuracy and scalability by aggregating diverse predictions, addressing the limitations of prior frameworks and enhancing VANET security.

## 3. Research questions

This study addresses the problem by answering the following research questions:

✓ What are the key characteristics and patterns of position falsification attacks in VANETs, and how can they be systematically classified?

✓ How does the proposed attack detection framework perform in detecting varied levels of attack types?

✓ How does the integration of bagging with machine learning models and hyper parameter tuning enhance the detection accuracy for position falsification attacks?

✓ How does this study further improve the robustness of detection?

## 4. Materials and methods

### 4.1. Overview of proposed research approach

This study investigates data-driven approaches for detecting position falsification attacks within Vehicular Ad Hoc Networks (VANETs) by employing machine learning to analyze Basic Safety Messages (BSMs) broadcast by vehicles. The approach utilizes the Bootstrap Aggregating (Bagging) technique alongside four classification models: Decision Trees [24], Random Forests [25], K-Nearest Neighbors (KNN) [26], and MLP classifiers [27]. The primary objective is to identify misbehavior in VANETs, focusing specifically on attacks where malicious entities falsify position data to undermine network integrity.

## 4.2. Proposed architecture

The proposed architecture for detecting position falsification attacks within Vehicular Ad Hoc Networks (VANETs) is built on a multi-layered framework that combines cryptographic security [28–30] and machine learning-based anomaly detection. At its core, the system consists of vehicles, Roadside Units (RSUs), and On-Board Units (OBUs), with an emphasis on minimizing computational overhead at the vehicle level. Vehicles periodically broadcast Basic Safety Messages (BSMs) that include crucial data such as position, speed, timestamp, and unique message identifiers. To ensure the authenticity of these messages, each vehicle is assigned a unique pair of public and private keys by a central authority. These keys are used to digitally sign the BSMs, ensuring that the data transmitted by vehicles is authentic and cannot be tampered with, forming a foundation for the machine learning detection system.

The detection process is carried out at RSUs, as shown in Fig 2, which collect BSMs from nearby vehicles and perform analysis on consecutive BSMs, comparing patterns in vehicle behavior to detect anomalies. This analysis leverages ensemble learning models, including Decision Trees, Random Forests, and K-Nearest Neighbors (KNN) with Bagging, to classify vehicles as either legitimate or potentially malicious. If any inconsistencies in the reported positions or other parameters are detected, such as deviations beyond predefined thresholds, the RSU flags the vehicle as a potential attacker. Once an attack is identified, the RSU broadcasts an alert to nearby vehicles and infrastructure nodes, warning them of the detected threat.

To optimize scalability and efficiency, the system offloads the computation required for data analysis to the RSUs, reducing the computational burden on the vehicles themselves. By offloading these tasks, the system ensures real-time detection and alerts without overburdening the vehicle's processing capabilities, making it well-suited for large-scale VANET environments.

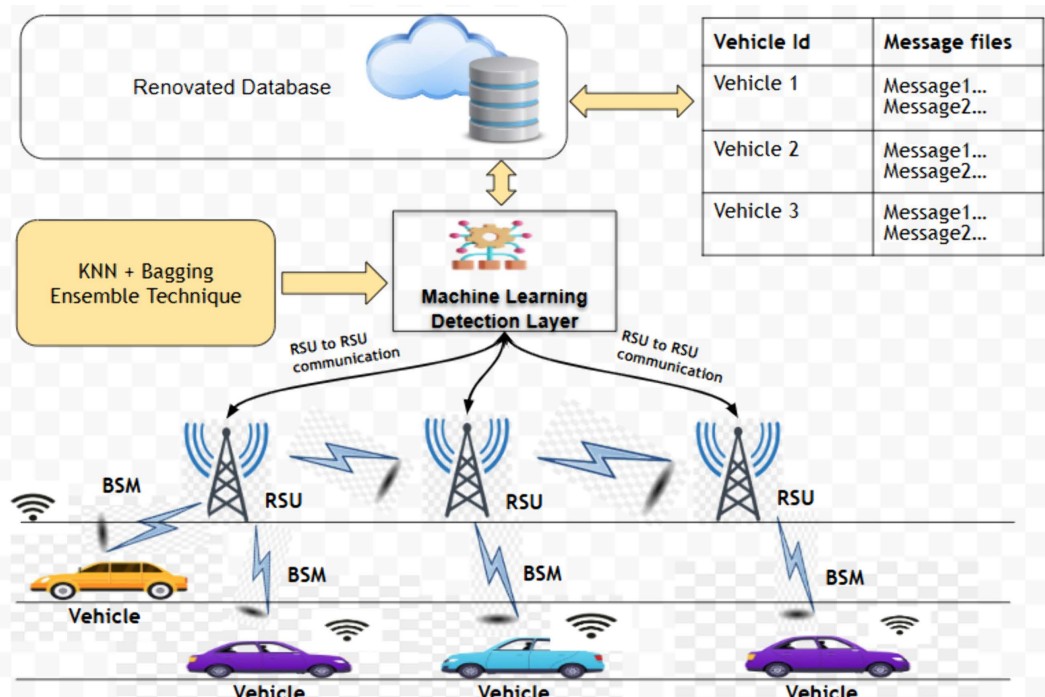

**Fig 2. Proposed architecture.**

## 4.3. Data extraction

The dataset used in this study is the VeReMi dataset [13], which consists of 225 simulation scenarios representing diverse traffic conditions (see Table 1) within Vehicular Ad-Hoc Networks (VANETs). The VeReMi dataset is generated using a combination of VEINS [31], SUMO, and OMNET++ simulation tools [32]. VEINS integrates SUMO, a tool that simulates urban mobility, with OMNET++, a network simulation framework, to create a realistic VANET environment. The Luxembourg traffic scenario (LuST) [13] is used to assess VANET applications within this dataset.

As Fig 3 indicates, each simulation contains the ground truth files and log files. The ground truth files provide the actual network behavior, including the identification of misbehaving and legitimate vehicles, as well as the types of attacks (such as position falsification). The log files, generated by each vehicle, record the BSMs received from other vehicles within the network. The data extraction process (see Fig 4) involves matching ground truth files with the corresponding log files from each simulation.

This process constructs a labeled dataset, categorizing each entry as either a legitimate or misbehaving vehicle. The VeReMi dataset includes several simulated attack types (Fig 5) such as constant position, constant offset

**Table 1. Simulation Parameters used in VeReMi dataset.**

| Parameters | Value | Description |
|---|---|---|
| Mobility | SUMO LuST | Luxembourg SUMO traffic |
| Simulation Area | 2300, 5400-6300, 6300 | Various road types |
| Simulation duration | 100s | Duration of simulation |
| Attacker probability | (0.1,0.2, 0.3) | Attacker probability in the network |
| Simulation start | (3, 5, 7) h | Control density |
| Signal interference model | Two-ray interference | Default signal propagation model in VEINS. |
| Obstacle shadowing | Simple | Default model used in VEINS. |
| Shadowing | Log-normal | Default signal attenuation model used in VEINS. |
| MAC implementation | 802.11p | Default MAC protocol used in VEINS. |
| Thermal power | −110 dBm | Default thermal noise level in VEINS. |
| Bit-rate | 6 Mbps | Default data transmission rate in VEINS. |
| Sensitivity | −89 dBm | Default sensitivity level for signal reception. |
| Antenna model | Monopole on roof | Default antenna configuration in VEINS. |
| Beaconing rate | 1 Hz | Default rate at which vehicles send a beacon. |

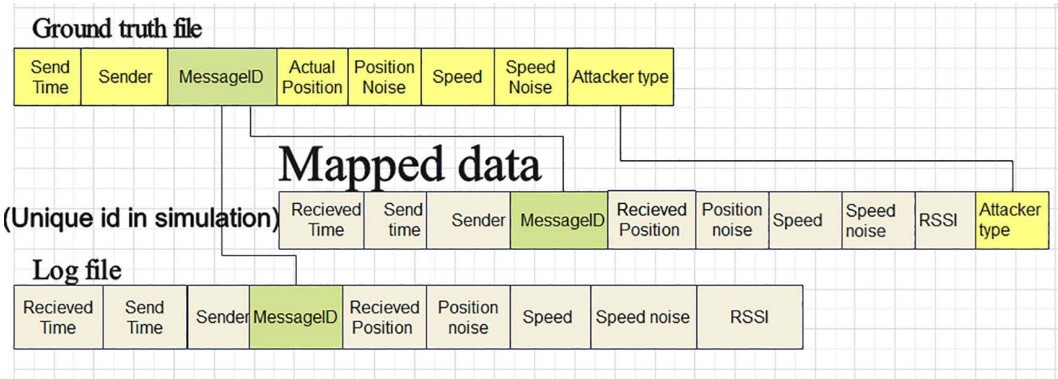

**Fig 3. Data extraction of Ground truth file and Log files [13].**

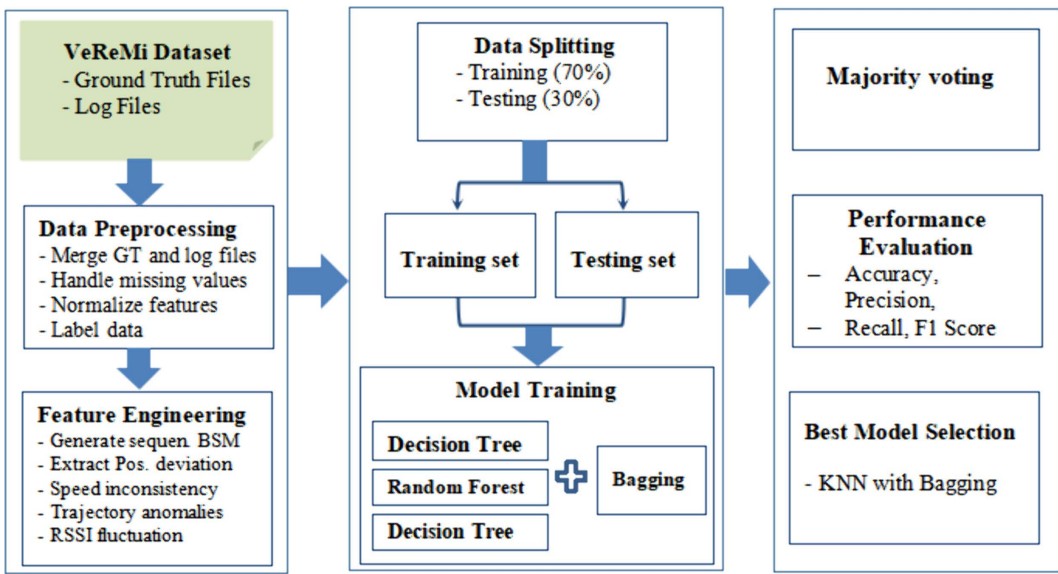

**Fig 4. Proposed methodology.**

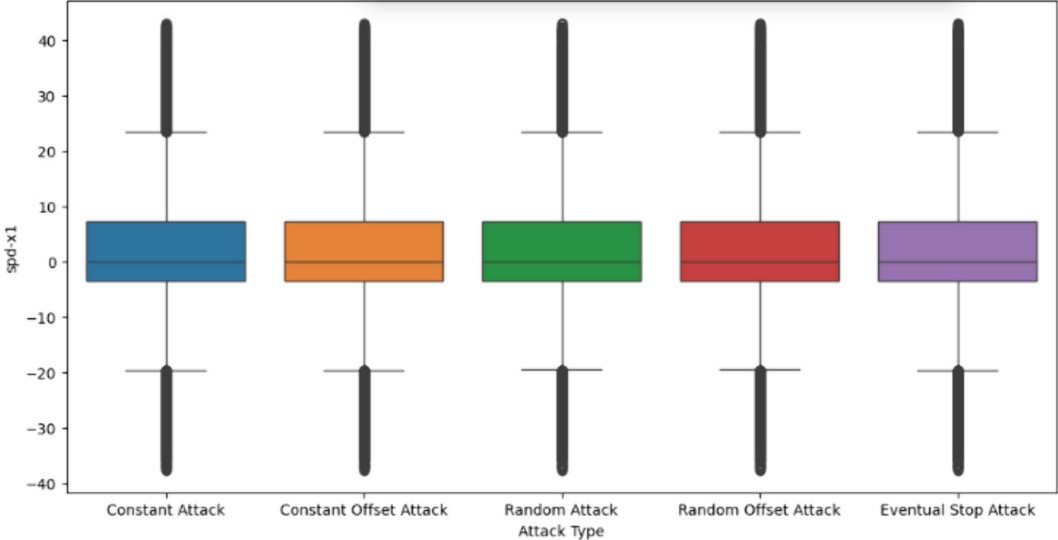

**Fig 5. Distribution of VeReMi datasets using box-plot for spd-x1.**

position, random position, random offset position, and eventual stop attacks [17,19,20,23] which are designed to evaluate the robustness of classification models against various position falsification scenarios, as summarized in Table 2.

**4.3.1. Feature importance analysis.** To enhance the interpretability of the KNN+Bagging model, we conducted a permutation importance analysis to quantify the contribution of each feature derived from the VeReMi dataset to the model's accuracy in detecting False Position Attacks (FPAs). The features include positional coordinates (pos-x1, pos-y1, pos-x2, pos-y2), speed components (spd-x1, spd-y1, spd-x2, spd-y2), and the time interval between consecutive Basic

**Table 2. Attack types and their description.**

| Attack Type | Attack Name | Attacker Behavior | Equations |
|---|---|---|---|
| Type-1 | Constant Position | The vehicle reports a fixed position throughout the attack. | Pnew = Pconst |
| Type-2 | Constant Offset Position | The attacker adds a constant offset to the vehicle's actual position before transmitting it. | Pnew = Pactual + Oconst |
| Type-4 | Random Position | The attacker sends a randomly chosen position, which can change unpredictably over time. | Pnew = Prando |
| Type-8 | Random Offset Position | A random offset is applied to the actual position, unpredictably altering the vehicle's location. | Pnew = Pactual + Orandom |
| Type-16 | Eventual Stop | The vehicle reports the same position for an extended period, simulating a sudden stop, then resumes normal movement. | (No specific equation) |

Safety Messages (BSMs) (time_interval, computed as sendtime_2 - sendtime_1). Permutation importance measures the decrease in model accuracy when a feature's values are randomly shuffled, using 10 repeats for computational efficiency.

As shown in in Table 3, pos-x2 and pos-y2 exhibit the highest importance across all attack types, with scores reaching 0.3732 and 0.3728 for Attack 4 (Constant Position) and Attack 8 (Random Offset Position), respectively. These features capture positional discrepancies critical to FPAs, reflecting the VeReMi dataset's emphasis on positional attack patterns. Earlier positional features (pos-x1, pos-y1) also contribute significantly, while speed features (spd-x1, spd-y1, spd-x2, spd-y2) are relevant only for Attack 16 (Eventual Stop), with scores up to 0.0408. The zero importance of time_interval likely results from fixed BSM intervals in the dataset, limiting temporal variability.

## 4.4. Classification models used

After preprocessing the VeReMi dataset, we proceed with dataset analysis and classification. This phase begins with the generation of a clean dataset by removing duplicates and filtering out irrelevant features. The dataset consists of consecutive Basic Safety Messages (BSMs) from each vehicle, with critical features like position and speed. The classification process begins by selecting machine learning models and tuning hyper parameters for optimal performance. We evaluate four models—Decision Tree, Random Forest, K-Nearest Neighbors (KNN), and Multilayer Perceptron (MLP)—each enhanced with Bagging (Bootstrap Aggregating). These models were chosen for their proven applications and ability to handle complex, noisy VANET data. Decision Trees are widely used in fraud detection for interpretability [24], Random Forests in network intrusion detection for robustness [25], KNN in VANET misbehavior detection for proximity-based accuracy [26], and MLPs in complex pattern recognition for modeling nonlinear relationships [27]. This study leverages these strengths, applying bagging to enhance detection accuracy and scalability in VANETs..

**4.4.1. Decision trees with bagging.** Decision trees are chosen for their simplicity, ease of interpretability, and ability to model non-linear relationships [24]. Their transparency allows for a clear understanding of how decisions are made, which is important for real-time decision-making in safety-critical systems like VANETs. The Gini impurity criterion is commonly used to determine the quality of a split:

**Table 3. Normalized feature importance score for KNN+bagging across attack types.**

| Attack Type | pos-x1 | pos-y1 | spd-x1 | spd-y1 | pos-x2 | pos-y2 | spd-x2 | spd-x2 | Time_ interval |
|---|---|---|---|---|---|---|---|---|---|
| Attack 1 | 0.2947 | 0.1820 | 0.0003 | 0.0002 | 0.2504 | 0.1652 | 0.0003 | 0.0002 | 0.0000 |
| Attack 2 | 0.2517 | 0.2325 | 0.0025 | 0.0036 | 0.2872 | 0.2425 | 0.0025 | 0.0035 | 0.0000 |
| Attack 4 | 0.3548 | 0.1539 | 0.0000 | 0.0000 | 0.3732 | 0.1618 | 0.0000 | 0.0000 | 0.0000 |
| Attack 8 | 0.2710 | 0.2704 | −0.0001 | −0.0001 | 0.3719 | 0.3728 | −0.0001 | −0.0001 | 0.0000 |
| Attack 16 | 0.2120 | 0.2250 | 0.0334 | 0.0304 | 0.2282 | 0.2525 | 0.0408 | 0.0341 | 0.0000 |

$$Gini(t) = 1 - \sum_{i=1}^{c} P^2_i \qquad (1)$$

where c is the number of classes, and Pi is the proportion of class i in node t. This criterion helps minimize the uncertainty at each split. However, while this model offers interpretability, it can suffer from overfitting when applied to noisy data, which is why we evaluate more advanced models.

**4.4.2. Random forest with bagging.** Random Forests, an ensemble method, are selected to address the potential overfitting issue of individual decision trees [25]. By aggregating predictions from multiple decision trees trained on different subsets of the data, Random Forests improve prediction accuracy and robustness. A key feature is random feature selection at each node:

$$BestSplit = argmax \left( \sum_{t \in T} Impurity(t) \right) \qquad (2)$$

where T represents the trees in the forest, and the best split is chosen based on minimizing impurity (e.g., Gini or Information Gain) over a random subset of features. This model also aggregates predictions using majority voting for classification:

$$\hat{y} = majority\ voting\ (\hat{y}1,\ \hat{y}2,\ \hat{y}3,\ \ldots, \hat{y}N) \qquad (3)$$

where y is the predicted label from the i$^{th}$ tree. They also handle high-dimensional data effectively, making them suitable for the complex and dynamic nature of VANETs. This model is chosen to increase model stability and enhance detection performance compared to single decision trees.

**4.4.3. K-nearest neighbors (KNN) with bagging.** KNN is a non-parametric model that classifies data based on the majority label of its closest neighbors, making it particularly effective for detecting local anomalies in vehicle behavior [26]. The Euclidean distance is commonly used to measure proximity:

$$d(x, x') = \sqrt{\sum_{i=1}^{n} (x_i - x'_i)^2} \qquad (4)$$

where x and x′ are two data points, and n is the number of features. By applying Bagging, we train multiple KNN classifiers on bootstrap samples and aggregate their predictions using majority voting:

$$y = \frac{1}{T} \sum_{t=1}^{T} sign \left( \sum_{K=1}^{K} wk.yk \right) \qquad (5)$$

where T is the number of KNN classifiers, wk is the weight of the kth nearest neighbor, and yk is the class label of the kth neighbor. This approach boosts generalization and reduces variance, making the model more robust to noisy data in VANETs.

**4.4.4. Multilayer Perceptron (MLP) Classifier with bagging.** Multilayer Perceptron (MLP), a powerful type of Artificial Neural Network (ANN), is selected for its ability to model complex non-linear relationships in the data [27]. The forward propagation step in MLP is defined as:

$$y^{(1)} = f \left( W^{(l)} x^{(l-1)} + b^{(l)} \right) \qquad (6)$$

where y(l) is the output of layer l, W(l) and b(l) are the weights and biases at layer l, x(l−1) is the input to layer l, and f is the activation function (e.g., ReLU, Sigmoid). During back-propagation, weights are updated based on the gradient of the loss function L, allowing the network to learn complex patterns in the data:

$$\frac{\partial L}{\partial W^{(l)}} = \frac{\partial L}{\partial y^{(l)}} \cdot \frac{\partial y^{(l)}}{\partial W^{(l)}}$$

(7)

Moreover, to elucidate the roles of the four bagging-enhanced classifiers in our vehicle-RSU pair-based framework, Table 4 provides a comparative analysis of Decision Tree, Random Forest, K-Nearest Neighbors (KNN), and Multilayer Perceptron (MLP), highlighting their application scenarios, similarities, and differences. This comparison underscores their suitability for detecting position falsification attacks in VANETs, with bagging improving robustness across all models, while KNN with Bagging excels due to its scalability and accuracy.

## 4.5. Hyperparameter tuning

To optimize the K-Nearest Neighbors (KNN) algorithm with Bagging, we fine-tuned hyperparameters to enhance detection of position falsification attacks in VANETs' dynamic, noisy environment [8]. The approach ensures robust performance through bagging, which reduces model variance by training multiple KNN instances on bootstrap samples and aggregating predictions via majority voting to mitigate noisy Basic Safety Messages (BSMs) [13]. Offloading computation to Roadside Units (RSUs) supports scalable, real-time processing (Section 4.2), while KNN's proximity-based anomaly detection, enhanced by bagging, achieves 100% accuracy for Attack 1 and 97.85% for Attack 16 (Section 5.3), outperforming single-classifier methods [17,19]. The number of nearest neighbors (n_neighbors) was set to 3 to strike a balance between accuracy and computational efficiency, enabling the model to detect subtle anomalies in position data without being overly sensitive to noise. The weights parameter was configured to 'distance', ensuring that closer neighbors had a greater influence on the classification decision, thereby improving accuracy by prioritizing more relevant data points. Additionally, an ensemble of 10 KNN classifiers (n_estimators = 10) was employed to enhance the model's stability and robustness, reducing variance and improving generalization by training multiple classifiers on different subsets of the data. These hyper-parameter choices were informed by preliminary experimental results, which demonstrated that the KNN with bagging model outperformed traditional algorithms like Decision Trees and Random Forests in terms of detection accuracy, robustness, and computational efficiency, making it particularly well-suited for detecting position falsification attacks in the noisy, dynamic environment of VANETs.

**Table 4. Comparative Analysis of Bagging-Enhanced Classification Models.**

| Model | Application Scenarios | Similarities | Differences |
|---|---|---|---|
| Decision Tree with Bagging | Fraud detection, VANET misbehavior detection [10]; ideal for interpretable rule-based anomaly detection | Uses bagging for robustness; handles noisy VANET data | Highly interpretable; prone to overfitting without bagging; lower complexity than MLP |
| Random Forest with Bagging | Network intrusion detection, VANET trajectory analysis [33]; suitable for high-dimensional data | Uses bagging for robustness; ensemble-based | More robust than Decision Tree due to random feature selection; moderate complexity |
| KNN with Bagging | VANET misbehavior detection, proximity-based anomaly detection [34]; effective for spatial data like BSM coordinates | Uses bagging for robustness; non-parametric | High accuracy in dynamic environments; computationally intensive without RSU offloading |
| MLP with Bagging | Complex pattern recognition, VANET attack classification [35]; excels in nonlinear relationships | Uses bagging for robustness; handles complex data | High computational cost; less interpretable than Decision Tree or Random Forest |

The preliminary experiments were conducted using a subset of the VeReMi dataset to determine optimal hyperparameters for KNN with Bagging, focusing on n_neighbors, weights, and n_estimators due to their critical impact on KNN's performance with spatial BSM data [12]. A grid search was performed over n_neighbors (1–10), weights ('uniform', 'distance'), and n_estimators (5–20), evaluating accuracy and F1 score via 5-fold cross-validation to ensure robustness. Results showed n_neighbors = 3 and weights = 'distance' maximized accuracy by effectively capturing local anomalies, while n_estimators = 10 balanced stability and computational efficiency. These findings guided the final hyperparameter selection, ensuring rational and reproducible choices for VANET misbehavior detection.

Furthermore, a 5-fold cross-validation process was implemented to ensure robust model evaluation. The dataset was divided into five subsets, with each fold serving as the test set once. Performance metrics (precision, recall, accuracy, and F1 score) were computed across all folds to assess generalization and mitigate overfitting. Hyperparameter tuning was conducted exclusively for KNN due to its high sensitivity to parameters like n_neighbors and weights, which are critical for accurate anomaly detection in VANETs' spatial data [12]. Decision Tree, Random Forest, and Multilayer Perceptron (MLP) leveraged bagging's inherent robustness, allowing standard configurations to perform effectively without extensive tuning.

---

**Algorithm 1:** Detection of false position in VANETs through bagging ensemble

```
1.  Input: BSMs (Vehicle ID, Timestamp, Position, Speed, Path, AttackType)
    Output: Detection of malicious or legitimate BSMs
2.  Preprocess_BSM_Data(BSMs)
    For each BSM in BSMs:
      Position= (Lat, Lon), Speed, Heading, Path
      Normalized Speed, Normalized Position
      Handle missing values: Impute or remove
3.  Feature_Engineering(BSMs)
    For each BSM:
      Velocity=ΔTime/ ΔPosition
      Directional Change (Heading)
      Distance from Predicted Path=[Posnpredicted–Posnactual]
      Time Interval=Timestamp_i–Timestamp_{i-1}
      Anomalous Pattern=|Predicted Position–Actual Position|
4.  Train_Machine_Learning_Models(Data, Labels)
    For each model M ∈ {DTC, RFC, KNN, MLP with Bagging}:
      Train model M with Data and Labels
5.  Detect_Position_Falsification(BSMs, Models)
    For each BSM:
      Classify using model M:
        PredictionM=M. Predict(Features)
      If Malicious Attack Detected:
        Flag vehicle as malicious
        Log event for investigation
      Else:
        Continue normal processing
6.  Return: Results (Malicious or Legitimate)
```

---

## 5. Results

This section presents the performance evaluation of the proposed machine learning models (Decision Tree, Random Forest, KNN, and MLP classifier) enhanced with Bagging for detecting position falsification attacks in VANETs. The results are visualized using performance metrics such as accuracy, precision, recall, and F1 score. The analysis highlights the effectiveness of each model across different attack scenarios.

### 5.1. Performance of Decision Tree with bagging

The Decision Tree algorithm, enhanced with bagging, demonstrated consistent performance in detecting position falsification attacks. Bagging stabilized predictions by training multiple Decision Trees on different subsets of the dataset and aggregating their outputs through majority voting. This model achieved near-perfect precision, recall, accuracy, and F1 scores for simpler attacks like Attack 1 and Attack 4, as shown in Fig 6. For moderately complex attacks such as Attack 2 and Attack 8, the model maintained strong performance, showcasing its adaptability to varied attack patterns.

However, for the most complex scenario, Attack 16, the model experienced a slight decline in performance (Fig 7), likely due to the intricate nature of eventual stop behavior. Despite this, the Decision Tree with bagging remains highly effective, underscoring its reliability in detecting position falsification attacks in dynamic and noisy VANET environments. The ensemble approach mitigates overfitting, enhances generalization, and ensures consistent performance, making it a valuable tool for VANET security.

### 5.2. Performance of Random Forest with bagging

The Random Forest algorithm, inherently an ensemble of Decision Trees, leveraged bootstrap sampling and random feature selection to enhance diversity and robustness. For Attack 1, the model delivered exceptional performance, achieving near-perfect precision, recall, accuracy, and F1 scores, as shown in Fig 8.

However, its performance declines significantly for more complex attacks, with Attack 2 showing moderate results (84.3235% accuracy) and Attack 4 achieving relatively strong metrics (94.1426% accuracy), as in Fig 8 and 9. The model struggles for highly complex attacks like Attack 8 (75.4103% accuracy) and Attack 16 (72.8280% accuracy), indicating challenges in handling intricate attack patterns such as random offsets and eventual stop behaviors. While Random Forest with bagging excels in simpler scenarios, its performance variability in complex attacks suggests limitations in generalizing to highly dynamic and noisy data.

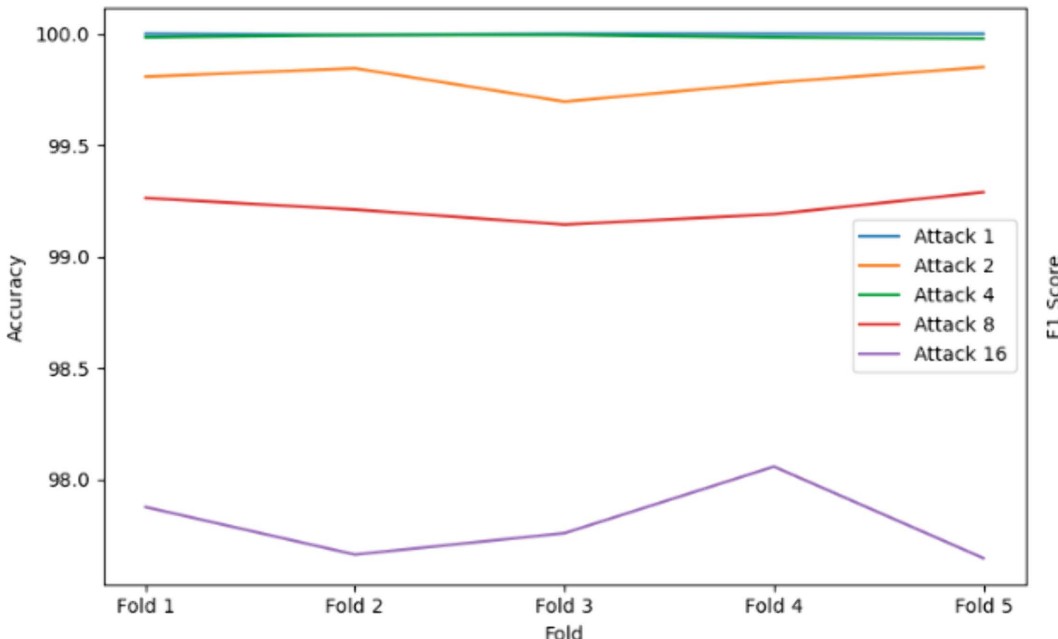

**Fig 6. Accuracy result using Decision Tree with Bagging.**

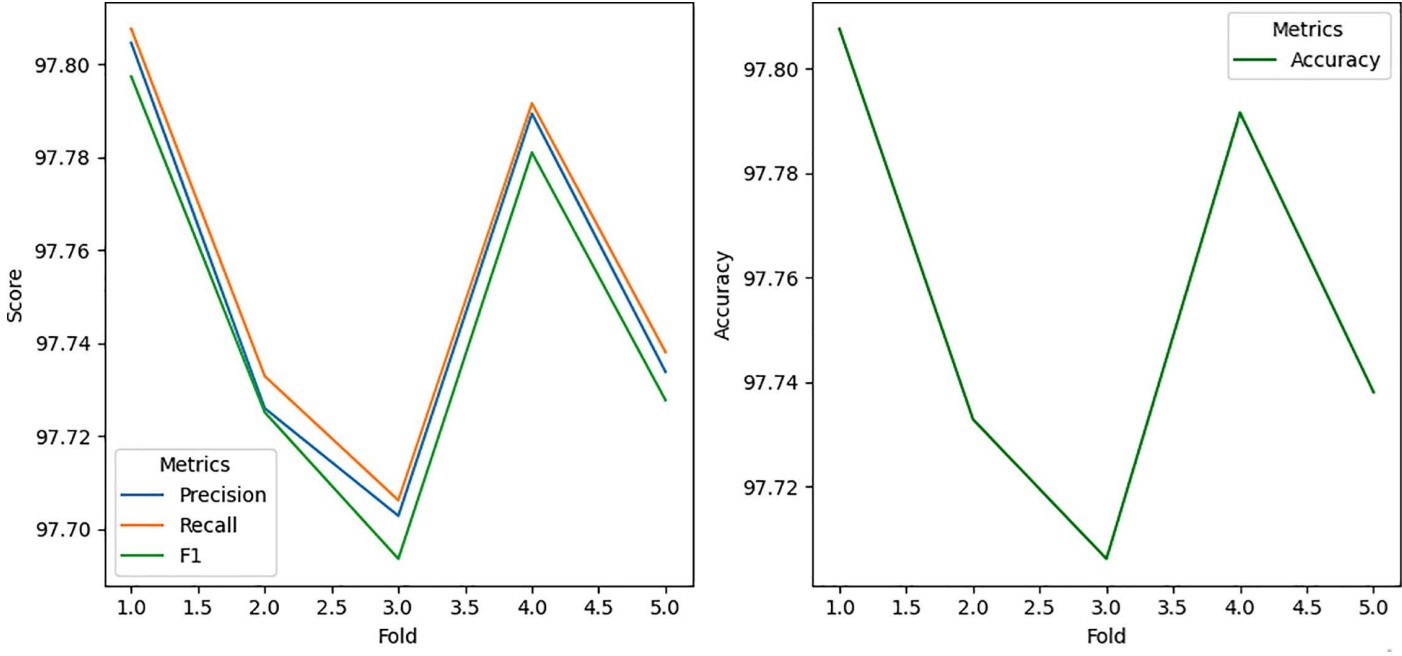

**Fig 7. Decision Tree with bagging performance for attack type 16.**

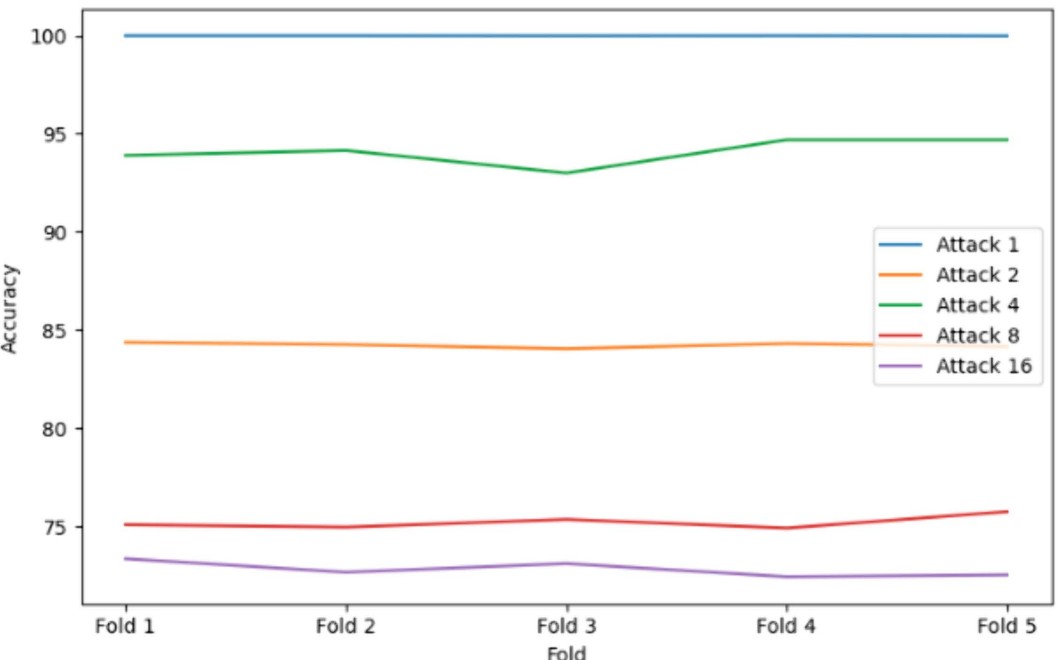

**Fig 8. Accuracy result using Random Forest Classifier with Bagging.**

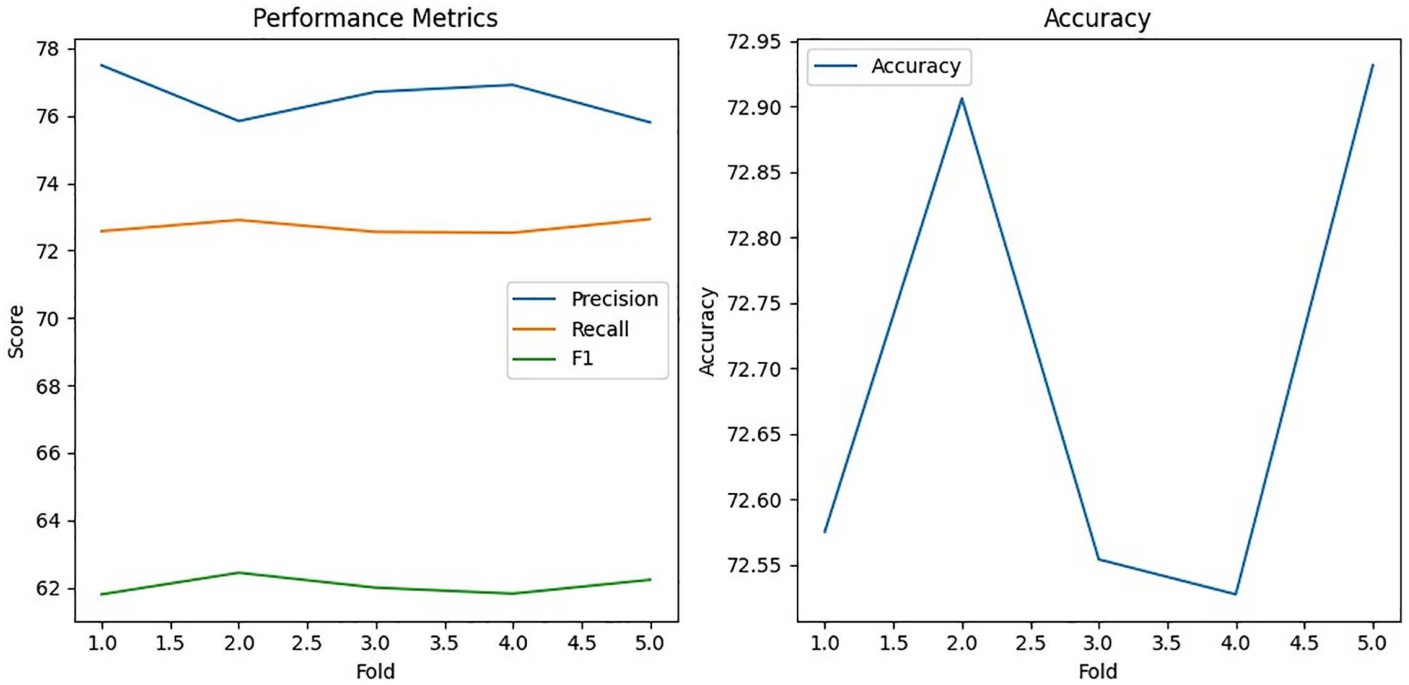

**Fig 9. Random Forest with bagging performance for attack type 16.**

### 5.3. Performance of K-Nearest Neighbors with bagging

The K-Nearest Neighbors (KNN) algorithm, integrated with Bagging, demonstrates outstanding performance across all attack scenarios, achieving perfect precision, recall, accuracy, and F1 scores (100%) for Attack 1, highlighting its exceptional ability to detect simple position falsification attacks. For Attack 2 and Attack 4, the model maintains near-perfect results, with accuracy exceeding 99.87% and 99.95% (Fig 10), respectively, showcasing its robustness in handling random and moderately complex attack patterns.

Even for more challenging scenarios like Attack 8 (99.1129% accuracy) and Attack 16 (97.8534% accuracy), as shown in Fig 11, KNN with Bagging performs remarkably well, with only a slight decline in performance, indicating its resilience to intricate attack behaviors such as random offsets and eventual stops. The ensemble model minimizes noise and captures intricate patterns in sequential BSMs, demonstrating its robustness and computational efficiency. This consistent high performance across all attack types underscores the effectiveness of KNN with bagging in detecting position falsification attacks in dynamic and noisy VANET environments, making it a highly reliable and scalable solution for enhancing VANET security with minimal need for further optimization.

### 5.4. Performance of MLP classifier with bagging

The MLP classifier with bagging shows mixed performance across attack types, achieving strong results for Attack 1 (95.2974% accuracy) and Attack 4 (99.9315% accuracy), indicating its effectiveness in detecting simpler and random position falsification attacks (Fig 12). However, its performance declines significantly for more complex attacks, with Attack 2 achieving moderate results (74.7137% accuracy) and Attack 8 showing relatively strong but inconsistent metrics (95.3923% accuracy).

For the most challenging scenario, Attack 16 (72.9593% accuracy), the model struggles, particularly in recall and F1 scores, highlighting its limitations in handling highly intricate attack patterns such as eventual stops as in Fig 13. While the

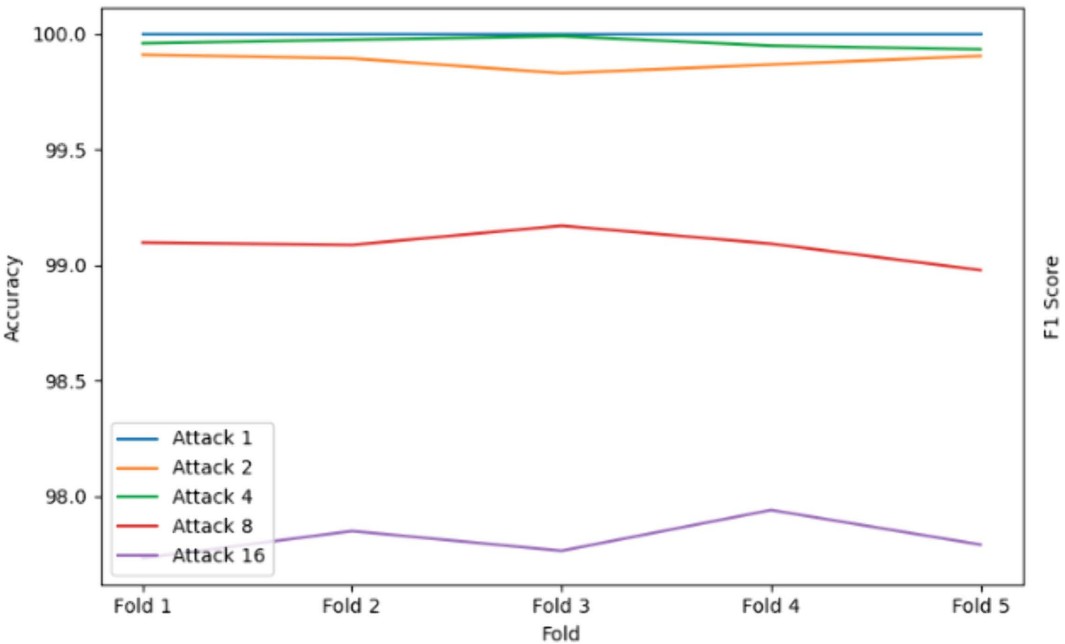

**Fig 10. Accuracy result using KNN Classifier with Bagging.**

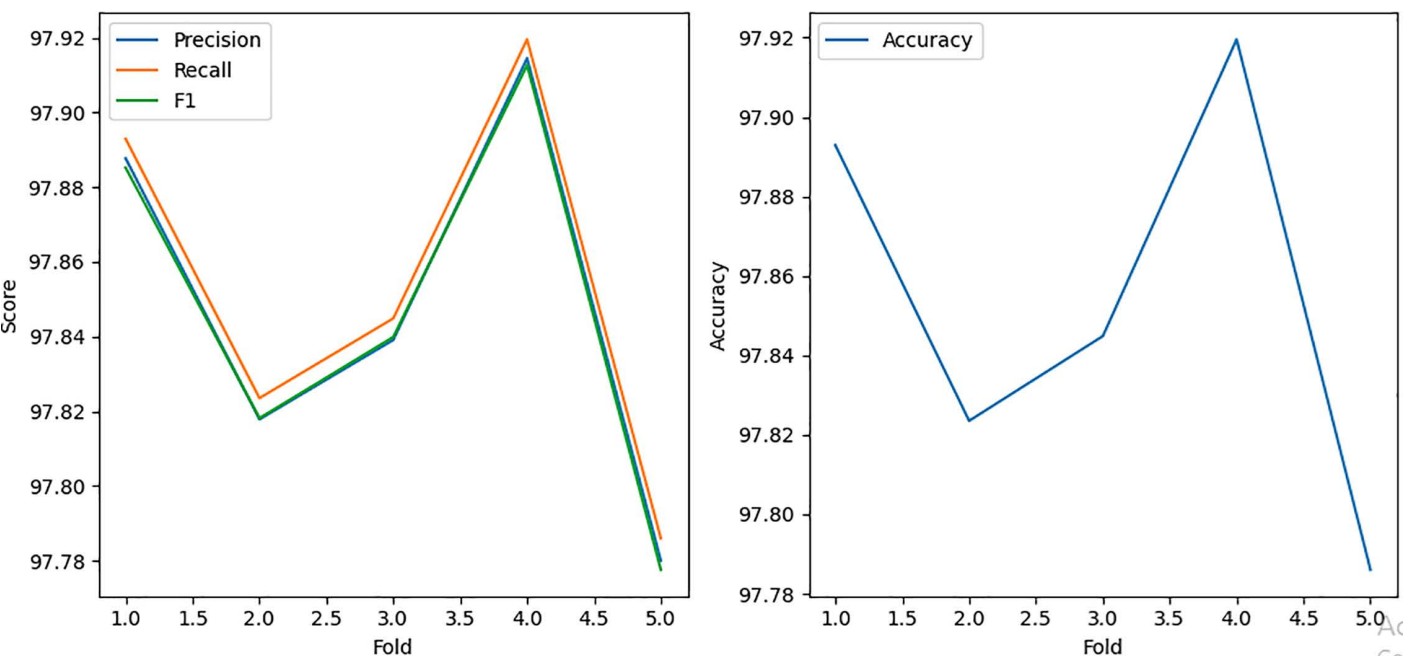

**Fig 11. KNN Classifier with bagging performance for attack type 16.**

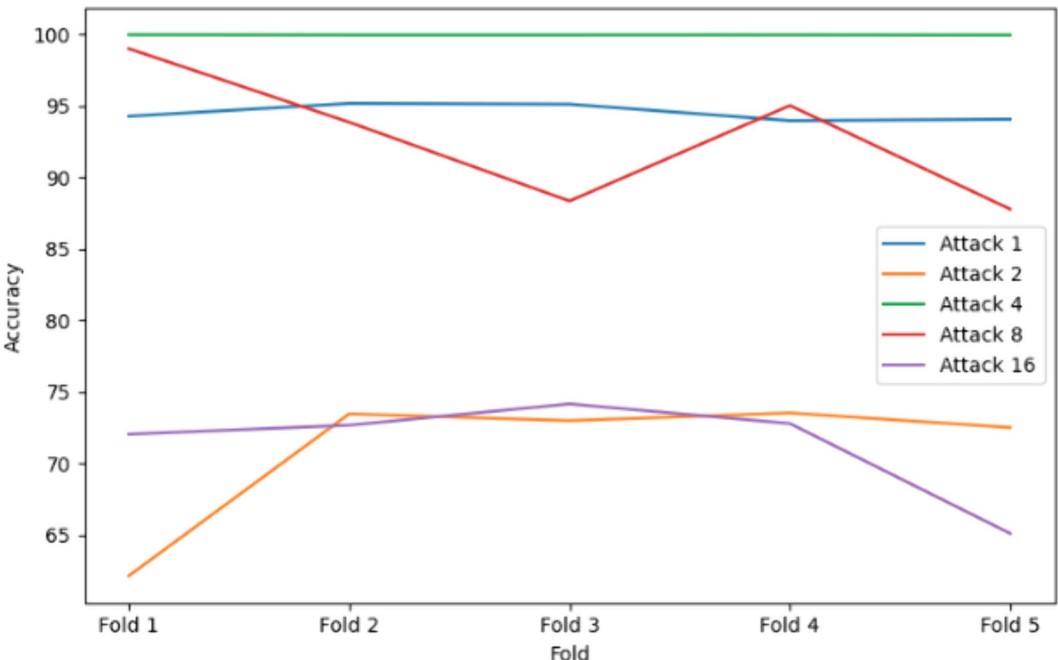

**Fig 12. Accuracy result using MLP Classifier with Bagging.**

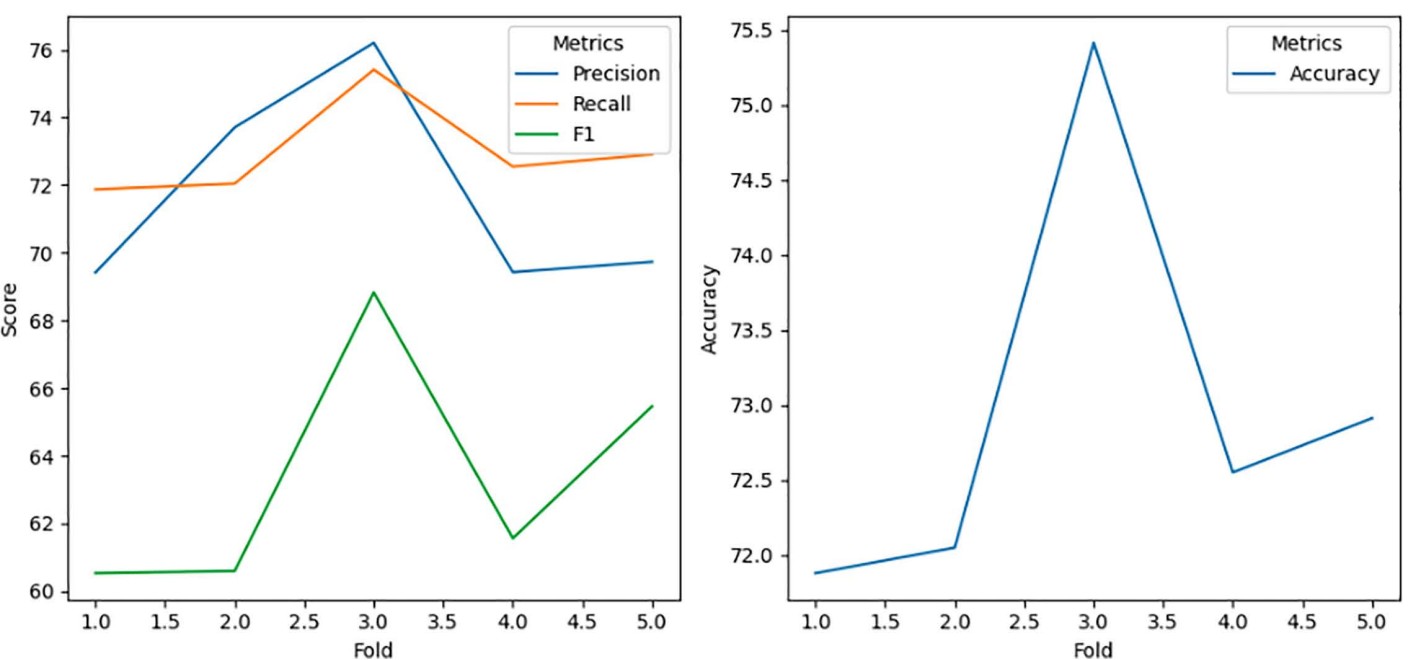

**Fig 13. MLP Classifier with bagging performance for attack type 16.**

MLP classifier with bagging performs well in simpler scenarios, its variability in complex attacks suggests challenges in generalizing to dynamic and noisy data.

### 5.5. Comparative analysis

Table 5 and Fig 14 highlight model performance across Attacks 1, 2, 4, 8, and 16. KNN with bagging consistently outperformed others, achieving perfect scores for Attack 1, near-perfect results for Attacks 2 (99.87%) and 4 (99.96%), and strong performance on Attacks 8 (99.11%) and 16 (97.85%). Its high recall is especially critical for minimizing false negatives in VANETs. Decision Tree with bagging was reliable across all attacks (>= 97.23%), though slightly impacted by Attack 16's complexity. Random Forest performed well in Attack 1 but showed reduced recall and accuracy in Attacks 8 and 16, suggesting challenges with complex patterns. MLP showed the most inconsistency—high accuracy in Attack 4 (99.9%) but poor recall in Attacks 2 (59.22%) and 16 (54.71%), limiting its reliability. Bagging and 5-fold cross-validation helped generalize performance, but only KNN maintained robustness across all scenarios, aided by tuned hyperparameters and effective proximity-based detection.

### 5.6. Generalization across Realistic Traffic scenarios

To evaluate the model's generalization beyond attack-specific characteristics, we simulated real-world traffic mobility scenarios. The data was segmented into two mobility regimes:

• Urban Scenario: BSMs where the vehicle speed < 10 km/h

• Highway Scenario: BSMs where the vehicle speed > 70 km/h

**Table 5. Performance of different ensemble model vs Attack types.**

| Attack Type | Algorithm | Mean Precision (%) | Mean Recall (%) | Mean Accuracy (%) | Mean F1 Score (%) |
|---|---|---|---|---|---|
| Attack 1 | Decision Tree + Bagging | 99.9989 | 99.9989 | 99.9919 | 99.9989 |
| | Random Forest + Bagging | 99.9989 | 99.9989 | 99.9989 | 99.9989 |
| | KNN + Bagging | 100.0 | 100.0 | 100.0 | 100.0 |
| | MLP Classifier + Bagging | 94.3219 | 97.2075 | 96.1852 | 95.5311 |
| Attack 2 | Decision Tree + Bagging | 99.7972 | 99.7973 | 99.6272 | 99.7971 |
| | Random Forest + Bagging | 94.2679 | 97.1399 | 94.2679 | 92.2042 |
| | KNN + Bagging | 99.8676 | 99.8676 | 99.8676 | 99.8676 |
| | MLP Classifier + Bagging | 75.6740 | 59.2161 | 74.6878 | 58.7610 |
| Attack 4 | Decision Tree with Bagging | 99.88 | 99.88 | 99.58 | 99.88 |
| | Random Forest with Bagging | 94.84 | 94.41 | 94.41 | 94.24 |
| | KNN with Bagging | 99.96 | 99.96 | 99.96 | 99.96 |
| | MLP Classifier + Bagging | 99.9265 | 99.8423 | 99.8996 | 99.8842 |
| Attack 8 | Decision Tree with Bagging | 97.24 | 97.24 | 97.24 | 97.23 |
| | Random Forest with Bagging | 91.78 | 95.35 | 95.35 | 98.89 |
| | KNN with Bagging | 99.13 | 99.11 | 99.11 | 99.11 |
| | MLP Classifier + Bagging | 97.2817 | 93.4186 | 95.9986 | 95.0803 |
| Attack 16 | Decision Tree with Bagging | 97.77 | 97.77 | 97.77 | 97.76 |
| | Random Forest with Bagging | 96.56 | 92.70 | 92.70 | 92.05 |
| | KNN with Bagging | 97.84 | 97.85 | 97.85 | 97.84 |
| | MLP Classifier + Bagging | 81.5867 | 54.7055 | 74.7217 | 51.4266 |

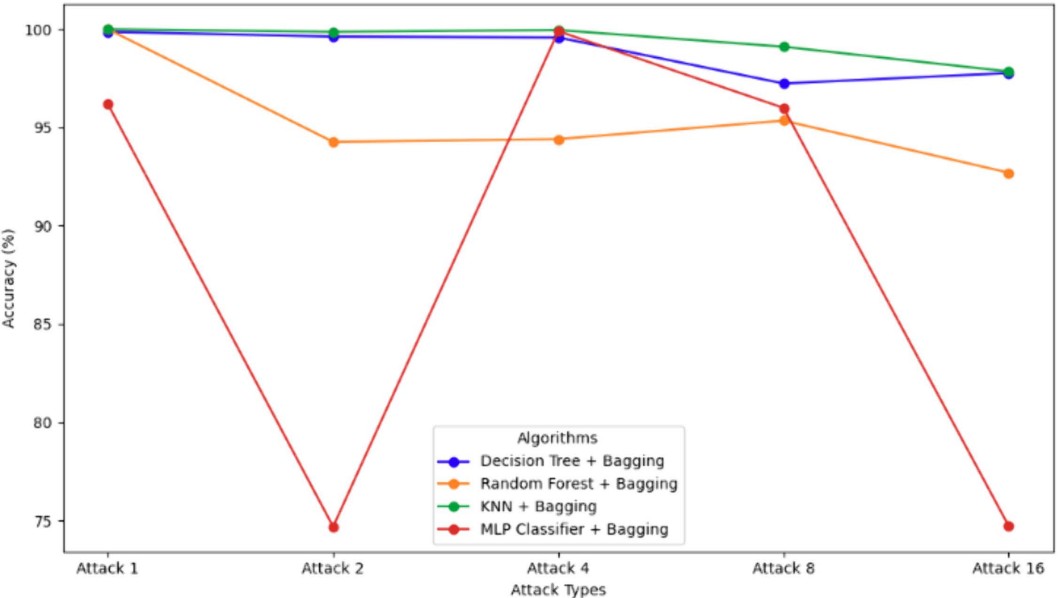

**Fig 14. Comparisons of Accuracy of the ensemble model vs Attack types.**

Each trained model was tested against all attack types (Attack 1, 2, 4, 8, and 16), as shown in Table 6, under both traffic conditions to assess its robustness across varying vehicle dynamics, such as stop-and-go congestion and high-speed movement.

The evaluation results, summarized in Table 6, reveal that KNN with Bagging consistently maintained high precision and F1-scores across both urban and highway environments. These findings support the model's ability to generalize beyond specific attack behaviors and adapt to diverse real-world traffic scenarios. Such simulation-based testing provides a valuable proxy in the absence of publicly available labeled real-world VANET datasets with falsified position attacks.

## 6. Discussion

The results confirm that ensemble learning techniques, specifically Bagging-based models, offer a robust approach for detecting False Position Attacks (FPAs) in VANETs. Among the algorithms tested, KNN + bagging consistently

**Table 6. Performance of KNN with bagging in urban vs. highway scenarios.**

| Attack Type | Scenario | Accuracy | Precision | Recall | F1-Score | Support |
|---|---|---|---|---|---|---|
| Type-1 | Urban | 1.00 | 1.00 | 1.00 | 1.00 | 7370 |
| Type-1 | Highway | 1.00 | 1.00 | 1.00 | 1.00 | 6436 |
| Type-2 | Urban | 1.00 | 1.00 | 1.00 | 1.00 | 7302 |
| Type-2 | Highway | 1.00 | 1.00 | 1.00 | 1.00 | 6459 |
| Type-4 | Urban | 1.00 | 0.99 | 1.00 | 0.99 | 7216 |
| Type-4 | Highway | 1.00 | 0.99 | 1.00 | 1.00 | 6432 |
| Type-8 | Urban | 0.97 | 0.99 | 0.91 | 0.95 | 7293 |
| Type-8 | Highway | 0.97 | 0.99 | 0.9 | 0.94 | 6526 |
| Type-16 | Urban | 0.97 | 0.99 | 0.89 | 0.94 | 7316 |
| Type-16 | Highway | 0.95 | 0.99 | 0.85 | 0.92 | 6566 |

outperformed others, almost in all attack types, achieving perfect scores across all metrics. This suggests that KNN, when combined with Bagging, is highly effective at handling the dynamic nature of FPAs in VANETs, likely due to its non-parametric and flexible nature. Both Decision Tree + bagging and Random Forest + bagging showed strong performance overall, with Decision Tree + bagging yielding consistent results across the board, particularly in Attack 2 and Attack 4. However, Random Forest + bagging experienced notable performance drops in more complex attacks such as Attack 8 and Attack 16, especially in recall and F1 score. This suggests that Random Forest may face challenges in certain attack scenarios, potentially due to its handling of feature diversity or overfitting in specific contexts.

On the other hand, the MLP Classifier + bagging demonstrated considerable variability across the attacks, illustrated in Fig 14. While it performed well in simpler attack types, it struggled with more complex attacks, as seen in Attack 2 and Attack 16, where recall and F1 scores were notably lower. This variability may be attributed to the model's sensitivity to complex decision boundaries in FPAs or its failure to capture certain attack-specific patterns. These findings highlight the need to select ensemble algorithms based on attack complexity, with KNN with bagging emerging as the most reliable for VANET security, followed by Decision Tree and Random Forest, while MLP requires further optimization to handle sophisticated attacks effectively.

While these models demonstrate strong performance in the VeReMi simulation, deploying the RSU-based detection system in real-world VANETs introduces practical constraints, such as communication interruptions and network latency, not modeled in VeReMi's idealized environment [13]. Interruptions, such as packet loss due to signal interference or urban obstacles, could result in missing BSMs, potentially reducing detection accuracy by providing incomplete positional data. Network latency, caused by RSU processing delays or high vehicle density, may hinder real-time alert generation, critical for safety-critical VANETs. Our system mitigates these risks through robust feature engineering (e.g., velocity, directional change) that tolerates minor data gaps and by offloading computation to RSUs, minimizing vehicle-side latency. However, significant interruptions or delays could impair performance, particularly for complex attacks like Attack 16.

## 7. Conclusion

The results of this study indicate that KNN with Bagging consistently outperforms the other algorithms in terms of accuracy and robustness across various attack scenarios. However, deploying such a system on a large scale requires further validation in real-world environments. It is crucial to test the system with a broader range of attack types and network conditions to evaluate its ability to generalize effectively. Future research should explore the integration of hybrid models, as well as other advanced machine learning techniques like deep learning or reinforcement learning, to enhance detection rates and counter adversarial learning, alongside simulations (e.g., using OMNET++) to address deployment constraints.

In conclusion, the findings highlight the significant potential of ensemble-based machine learning approaches, especially KNN with Bagging, in enhancing the security and resilience of VANETs against position falsification attacks. The successful application of these models at the Roadside Unit (RSU) level could provide a robust framework for real-time attack detection and mitigation, contributing to the broader objective of creating secure and efficient Intelligent Transportation Systems (ITS). As such, this research lays a strong foundation for future advancements in VANET security.

## Acknowledgments

The authors would like to thank the Jimma Institute of Technology for supporting them through different resources. The authors would like to thank Jimma University for its support during the research work.

## Author contributions

**Conceptualization:** Bekan Kitaw Mekonen, Lemi Bane, Negasa Berhanu Fite.

**Data curation:** Bekan Kitaw Mekonen, Lemi Bane.

**Formal analysis:** Bekan Kitaw Mekonen, Lemi Bane, Negasa Berhanu Fite.

**Investigation:** Bekan Kitaw Mekonen, Lemi Bane.

**Methodology:** Bekan Kitaw Mekonen, Lemi Bane.

**Project administration:** Bekan Kitaw Mekonen, Lemi Bane.

**Resources:** Bekan Kitaw Mekonen, Lemi Bane, Negasa Berhanu Fite.

**Software:** Bekan Kitaw Mekonen, Lemi Bane.

**Supervision:** Bekan Kitaw Mekonen, Lemi Bane, Negasa Berhanu Fite.

**Validation:** Bekan Kitaw Mekonen, Lemi Bane.

**Visualization:** Bekan Kitaw Mekonen, Lemi Bane.

**Writing – original draft:** Bekan Kitaw Mekonen, Lemi Bane.

**Writing – review & editing:** Bekan Kitaw Mekonen, Lemi Bane, Negasa Berhanu Fite.

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
