## [Decision Letter · Decision Letter 0]

Dear Dr. Mekonen,

Thank you for submitting your manuscript to PLOS ONE. After careful consideration, we feel that it has merit but does not fully meet PLOS ONE’s publication criteria as it currently stands. Therefore, we invite you to submit a revised version of the manuscript that addresses the points raised during the review process.

Dear authors, please revise the manuscript according to the comments

We look forward to receiving your revised manuscript.

Kind regards,

Gen Li

Academic Editor

PLOS ONE

3. Thank you for uploading your study's underlying data set. Unfortunately, the repository you have noted in your Data Availability statement does not qualify as an acceptable data repository according to PLOS's standards.

Additional Editor Comments:

Dear authors, please revise the manuscript according to the comments

Reviewers' comments:

Reviewer's Responses to Questions

**Comments to the Author**

1. Is the manuscript technically sound, and do the data support the conclusions?

Reviewer #1: Yes

Reviewer #2: Yes

2. Has the statistical analysis been performed appropriately and rigorously?

Reviewer #1: Yes

Reviewer #2: Yes

3. Have the authors made all data underlying the findings in their manuscript fully available?

Reviewer #1: Yes

Reviewer #2: Yes

4. Is the manuscript presented in an intelligible fashion and written in standard English?

Reviewer #1: Yes

Reviewer #2: Yes

Reviewer #1: This paper investigates the challenge of False Position Attacks (FPAs) in Vehicular Ad-hoc Networks (VANETs). By integrating ensemble learning techniques (specifically Bootstrap Aggregating) into the detection framework, the proposed approach enhances detection accuracy and system robustness. This methodology effectively addresses the dual limitations of cryptographic solutions in authenticating data credibility and conventional machine learning models' insufficient generalization capacity when confronting sophisticated attack patterns in dynamic vehicular environments. The following recommendations are provided for the author's consideration:

1.The experiments are solely based on VeReMi simulation data and do not validate performance in real-world traffic scenarios (e.g., urban congestion, high-speed mobility). Could real-road data testing or comparisons with public VANET real-world datasets (e.g., CarTel) be added to verify the generalization capability of the proposed model?

2.The experiments only cover five predefined attack types (e.g., fixed-position and random-offset attacks) and do not address more stealthy attacks (e.g., dynamic-offset or coordinated attacks). Due to insufficient coverage of attack types, could the attack scenarios be expanded—for instance, by simulating multi-node coordinated position spoofing or dynamically adjusted attack strategies—to evaluate the model’s capability in handling unknown attacks?

3.The "black-box" nature of KNN+Bagging may affect trustworthiness in practical deployment, and the paper lacks analysis on the contributions of critical features. It is recommended to further supplement the analysis in this aspect.

4.The experimental comparisons are limited to traditional machine learning models and do not include some latest advanced methods, such as Graph Neural Networks (GNN) and Federated Learning (FL). Could more comparative experiments or references to recent literature analyses be added to verify the superiority of the proposed method?

5.The study does not consider the impact of practical constraints in real-world deployment, such as communication interruptions and network latency between RSUs (Roadside Units) and vehicles, on the proposed detection system. It is suggested that the authors discuss this aspect.

Reviewer #2: This paper proposes an ensemble learning framework that uses CART, Random Forest, KNN, and MLP classifiers with bagging for Vehicular Ad-hoc Networks vulnerable to location forgery attacks to detect such attacks. By analyzing sequential BSMs from the VeReMi dataset via an RSU-level system, they get the results that KNN with bagging achieved 100% precision, recall, accuracy, and F1 score for Attack 1 and near-perfect performance for complex attacks. Other models have varying performance across simple and complex scenarios. The framework proves the effectiveness of ensemble methods, particularly KNN with bagging, in securing VANETs. The following questions are discussed with the author.

1. It is recommended to introduce the Public Key Infrastructure (PKI) model in the Introduction part before explaining its advantages and functions relative to Intelligent Transportation Systems (ITS).

2. The connection between the first and second paragraphs of the article is not strong. It is recommended to add content to smoothly connect the first paragraph, which emphasizes the necessity of a bad behavior detection model, with the content in the second paragraph regarding the security threats faced by Vehicular Ad Hoc Networks (VANETs).

3. The Introduction part and the Rationale part of the article have introduced the methods used in this article many times. The order of the methods lacks a logical connection, the transition is rigid, and a clear theoretical system is not formed. To enhance the readability and academic standardization of the article, it is suggested that the author should reorganize the chapter structure of the method, integrate the repetitive content, adopt the method of comparative analysis or progressive logic arrangement, simplify the expression, highlight the core technical details and innovation points.

4. It is recommended to introduce the existing applications of the four classification models used in this study, such as Decision tree, Random Forest, K-Nearest Neighbor algorithm and Multilayer Perceptron classifier, so that readers can better understand the research mechanism of this study.

5. Because the method proposed in this paper aims to further improve the accuracy and robustness of detection, add the following research question to solve the problem of bad behavior detection in vehicular ad hoc networks (VANETs): "How does this study further improve the robustness of detection?" And add the relevant content in the article to explain it.

6. It is recommended to add a comparative analysis of the four machine learning models combined with the bagging method in the fourth part of the article, presented in the form of tables, so that their respective application scenarios, similarities and differences are more intuitive.

7. In this paper, the bagging method is combined with four machine learning models, but only the hyperparameter tuning of KNN is performed. The author explains the reason for choosing KNN in the paper.

8. In this paper, the selection of hyperparameters is based on the preliminary experimental results. However, to make readers better understand the basis and rationality of the selection of hyperparameters, it is suggested that the author should increase the principle and process of the preliminary experiment, so that other researchers can refer to and learn from similar work.

9. In the research, although accuracy, recall, precision and F1-score are proposed as the evaluation indexes of the generalization ability of the model, the specific data of accuracy, recall, precision and F1 - score are not elaborated and analyzed in the text, which leads to the absence of key data in the evaluation system and affects the integrity and objectivity of the model performance evaluation.

**Do you want your identity to be public for this peer review?** For information about this choice, including consent withdrawal, please see our Privacy Policy

Reviewer #1: No

Reviewer #2: No

---

## [Author Response · Author response to Decision Letter 1]

15 Jun 2025

We would like to express our gratitude for your valuable feedback. We have addressed each of your comments and made the necessary revisions to improve the manuscript.

---

## [Decision Letter · Decision Letter 1]

Detection of false position attacks in VANETs through bagging ensemble learning

PONE-D-25-17735R1

Dear Dr. Mekonen,

We’re pleased to inform you that your manuscript has been judged scientifically suitable for publication and will be formally accepted for publication once it meets all outstanding technical requirements.

Kind regards,

Gen Li

Academic Editor

PLOS ONE

Additional Editor Comments (optional):

Reviewers' comments:

Reviewer's Responses to Questions

**Comments to the Author**

Reviewer #1: All comments have been addressed

Reviewer #2: All comments have been addressed

2. Is the manuscript technically sound, and do the data support the conclusions?

Reviewer #1: Yes

Reviewer #2: Yes

3. Has the statistical analysis been performed appropriately and rigorously?

Reviewer #1: Yes

Reviewer #2: Yes

4. Have the authors made all data underlying the findings in their manuscript fully available?

Reviewer #1: Yes

Reviewer #2: Yes

5. Is the manuscript presented in an intelligible fashion and written in standard English?

Reviewer #1: Yes

Reviewer #2: Yes

Reviewer #1: (No Response)

Reviewer #2: Vehicle mounted ad hoc networks (VANETs) are crucial for intelligent transportation systems (ITS), as they enable vehicle to vehicle (V2V) and vehicle to infrastructure (V2I) communication to improve road safety and traffic flow. The proposed framework emphasizes the effectiveness of integration technology, especially KNN with bagging, in protecting VANET communication systems, providing a scalable, efficient, and robust solution for VANET security. The paper has clear ideas, concise methods, and reliable conclusions. It is recommended to accept this manuscript.

**Do you want your identity to be public for this peer review?** For information about this choice, including consent withdrawal, please see our Privacy Policy

Reviewer #1: No

Reviewer #2: **Yes: ** Baohua Guo

---

## [Editor Report · Acceptance letter]

PONE-D-25-17735R1

PLOS ONE

Dear Dr. Mekonen,

I'm pleased to inform you that your manuscript has been deemed suitable for publication in PLOS ONE. Congratulations! Your manuscript is now being handed over to our production team.

Kind regards,

on behalf of

Dr. Gen Li

Academic Editor

PLOS ONE